# A Metagenomic Investigation of Spatial and Temporal Changes in Sewage Microbiomes across a University Campus

Noah Fierer,[a,b] Hannah Holland-Moritz,[c] ⓘ Alexandra Alexiev,[a] Harpreet Batther,[d] Nicholas B. Dragone,[a,b] Liam Friar,[d] Matthew J. Gebert,[a,b] Sarah Gering,[a,b] Jessica B. Henley,[b] Sierra Jech,[a] Emily M. Kibby,[e] Tina Melie,[a] William B. Patterson,[f] Eric Peterson,[g] Kyle Schutz,[a] Elías Stallard-Olivera,[a,b] John Sterrett,[f] Corinne Walsh,[a,b] Cresten Mansfeldt[g]

[a]Department of Ecology and Evolutionary Biology, University of Colorado, Boulder, Colorado, USA
[b]Cooperative Institute for Research in Environmental Sciences, University of Colorado, Boulder, Colorado, USA
[c]Department of Natural Resources and the Environment, University of New Hampshire, Durham, New Hampshire, USA
[d]Department of Geological Sciences University of Colorado, Boulder, Colorado, USA
[e]Department of Biochemistry, University of Colorado, Boulder, Colorado, USA
[f]Department of Integrative Physiology, University of Colorado, Boulder, Colorado, USA
[g]Department of Civil, Environmental, and Architectural Engineering, University of Colorado, Boulder, Colorado, USA

**ABSTRACT** Wastewater microbial communities are not static and can vary significantly across time and space, but this variation and the factors driving the observed spatiotemporal variation often remain undetermined. We used a shotgun metagenomic approach to investigate changes in wastewater microbial communities across 17 locations in a sewer network, with samples collected from each location over a 3-week period. Fecal material-derived bacteria constituted a relatively small fraction of the taxa found in the collected samples, highlighting the importance of environmental sources to the sewage microbiome. The prokaryotic communities were highly variable in composition depending on the location within the sampling network, and this spatial variation was most strongly associated with location-specific differences in sewage pH. However, we also observed substantial temporal variation in the composition of the prokaryotic communities at individual locations. This temporal variation was asynchronous across sampling locations, emphasizing the importance of independently considering both spatial and temporal variation when assessing the wastewater microbiome. The spatiotemporal patterns in viral community composition closely tracked those of the prokaryotic communities, allowing us to putatively identify the bacterial hosts of some of the dominant viruses in these systems. Finally, we found that antibiotic resistance gene profiles also exhibit a high degree of spatiotemporal variability, with most of these genes unlikely to be derived from fecal bacteria. Together, these results emphasize the dynamic nature of the wastewater microbiome, the challenges associated with studying these systems, and the utility of metagenomic approaches for building a multifaceted understanding of these microbial communities and their functional attributes.

**IMPORTANCE** Sewage systems harbor extensive microbial diversity, including microbes derived from both human and environmental sources. Studies of the sewage microbiome are useful for monitoring public health and the health of our infrastructure, but the sewage microbiome can be highly variable in ways that are often unresolved. We sequenced DNA recovered from wastewater samples collected over a 3-week period at 17 locations in a single sewer system to determine how these communities vary across time and space. Most of the wastewater bacteria, and the antibiotic resistance genes they harbor, were not derived from human feces, but human usage patterns did impact how the amounts and types of bacteria and bacterial genes we found in these systems varied over time. Likewise, the wastewater communities, including both bacteria and their viruses, varied depending on location within the sewage network, highlighting the challenges and opportunities in efforts to monitor and understand the sewage microbiome.

Address correspondence to Noah Fierer, noahfierer@gmail.com.

The authors declare no conflict of interest.

**KEYWORDS** antibiotic resistance genes, bacteriophage, metagenomics, sewage, sewer systems, wastewater

While often hidden from view and rarely attracting our attention (unless they malfunction), municipal sewage systems are important microbial ecosystems. Domestic wastewater harbors large numbers of microorganisms, including bacteria, archaea, protists, and viruses. The microbial taxa found in these wastewaters can be derived from human sources (including fecal, skin, and oral microbes deposited into the system) and external environmental sources (including soil, groundwater, or sediment microbes introduced during passage through the system and microbes introduced directly from tap water) (1). In addition to these more transient inputs, many of the microbes found in wastewater streams are more permanent residents of sewage systems found in the sewer sediments and biofilms (1).

Research into the microbiology of sewage systems can provide valuable insights relevant to a range of basic and applied scientific disciplines, including epidemiology, microbial ecology, environmental engineering, and human microbiome research. For example, previous work has demonstrated that studies of microbes found in sewage can be used to track pathogen prevalence in a population (2), understand pipe corrosion (3–5), track the presence and dissemination of antibiotic resistance genes (6–8), characterize fecal microbiomes across human populations (9), and contribute to an understanding of the biogeochemical processes occurring in wastewater (10, 11).

Sewer systems harbor dynamic microbial communities; there is no single type of wastewater microbial community. Rather, like other microbial ecosystems, the microbial communities found in sewage systems can be highly variable across time and space. This is true regardless of the specific aspect of wastewater microbial communities studied. The composition of bacterial, archaeal, viral, and microeukaryotic communities in wastewater communities can exhibit pronounced variation at time scales ranging from hours to months (12–14) with appreciable spatial variation within a given sewage network (13) or across different geographic regions (9, 15). The same goes for particular genes or gene categories of interest (including antibiotic resistance genes) which can also vary with respect to their diversity and abundances depending on the sampling location and the timing of sample collection (8, 16, 17).

From those studies that have comprehensively assessed spatial and temporal variation in the wastewater microbiome, we know that a number of factors can influence the amounts and types of microbes found in wastewater. These factors can include temperature, position along a wastewater system, the human populations contributing to a given wastewater system, system materials, or other characteristics (13). This potential for high spatial and temporal variation underlies much of the promise of using wastewater for the epidemiological surveillance of human pathogens (2, 18, 19). To use a recent example, there are many ongoing and published studies focused on detecting SARS-CoV-2 in wastewater samples to detect changes in the prevalence of COVID-19 infections over time or across locations (20, 21). This potential for a high degree of spatial and temporal variation in microbial distributions is not restricted to specific pathogens but can also be observed when looking more broadly at the taxonomic structure of sewage-associated communities (9, 12, 13) and their genomic attributes, as evidenced from recent work using shotgun metagenomic approaches to study wastewater microbial communities (15, 16, 22). Quantifying the spatial and temporal variation in wastewater microbial communities is critical for understanding the factors that influence these microbial communities and their functional capabilities, identifying the likely sources of particular microorganisms (or microbial genes) relevant to public health, monitoring the impacts of changes in the design and operation of sewage networks, and understanding the potential for specific biogeochemical processes to occur in sewage systems, including those associated with nutrient cycling and biocorrosion (1). The value of documenting spatial and temporal variation is not restricted to the study of wastewater microbial communities, but their dynamic nature, their

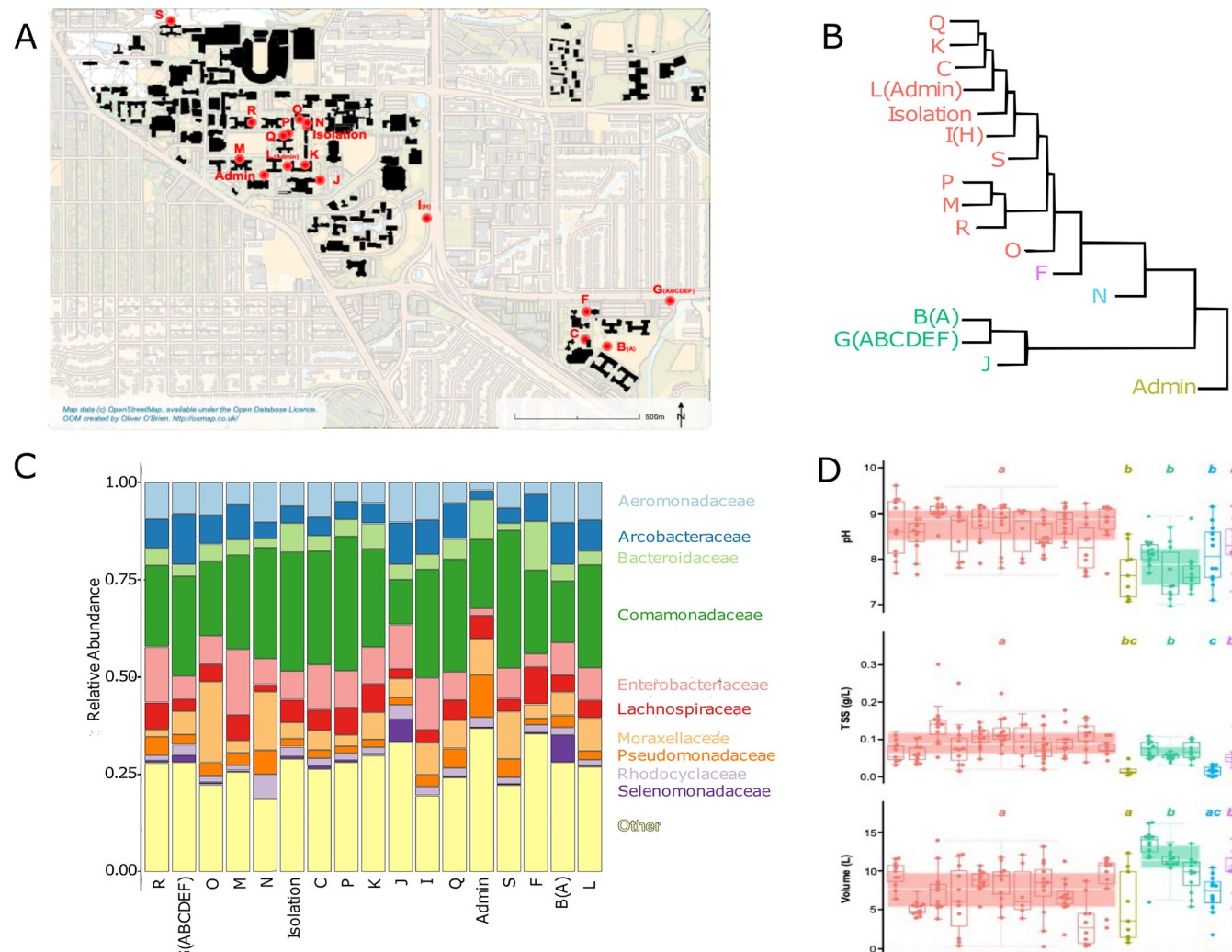

**FIG 1** (A) Map of the 17 locations across the University of Colorado campus from which sewage samples were collected for this study with each sampling location labeled following t naming convention used previously (50). (B) Summary presentation of the identified clusters of community types across the 17 sampling locations based on the overall degree of dissimilarity in community composition, with results shown using a hierarchical clustering diagram (Ward method), as calculated from average Bray-Curtis dissimilarity scores per location. (C) Average relative abundances (proportions) of the dominant bacterial families across each of the 17 locations. (D) Variation in the three measured variables significantly associated with spatial differences in community composition, across each location and across the five identified clusters. Note that panels B to C do not highlight the temporal variation observed within individual locations since these are just based on average dissimilarities in community composition (panel B) and mean relative abundances within a given sampling location (panel C).

direct and indirect associations with human populations served by sewer systems, and their potential relevance to public health make studies of the "sewer microbiome" broadly relevant.

Despite the large body of previous work on the microbiology of sewage systems, what often remains undetermined is how spatial and temporal variation in the structure of microbial communities and their genomic attributes compares at finer levels of spatiotemporal resolution, i.e., the variation across multiple locations within a single sewage network over days to weeks. Our study was designed to address this knowledge gap. We collected 188 wastewater samples from 17 locations on the University of Colorado campus, with each location representing wastewater outflow from a single building, or cluster of buildings, within the same sewer network (Fig. 1A). Samples were collected from each of the locations up to four times per week for 3 weeks. This sampling intensity allowed us to identify spatial patterns across a sewage network, assess the degree to which the wastewater communities exhibit shared temporal dynamics across the network, and determine

whether these spatiotemporal patterns are predictably related to differences in building occupancy, wastewater chemistry, sewer construction, and weather events. Each sample was analyzed using a shotgun metagenomic approach, with the resulting data used to explore spatiotemporal patterns in bacterial and archaeal community composition and their functional genes, with a focus on antibiotic resistance genes given their relevance to public health (23, 24). We also used these data to assess spatiotemporal patterns in viral (phage) communities as they have potentially important, but understudied, contributions to the biology of these systems (15), testing the hypothesis that changes in viral communities would mirror the associated changes in host microbial communities, as has been observed in other systems (25–27). As one of the more comprehensive studies of the wastewater microbiome within a single system, this work highlights the utility of understanding spatiotemporal dynamics in these microbial communities.

## RESULTS AND DISCUSSION

**General description of the prokaryotic communities.** Across the entire data set, we identified 1,087 unique prokaryotic taxa using phyloFlash (28). Since eukaryotic taxa represented <0.3% of extracted rRNA gene reads, we focus on the prokaryotic communities here. On average, we identified 357 unique prokaryotic taxa per sample (range, 182 to 530), but nearly all of these were bacterial. We extracted rRNA genes from seven archaeal taxa (all members of the *Methanobacteriales*, *Methanomicrobiales*, and *Methanosarcinales* groups), and these archaeal taxa represented only 0.02% of the rRNA gene reads (range, 0 to 0.4% per sample). The bacterial communities were dominated by members of the following phyla: *Proteobacteria*, *Firmicutes*, *Bacteroidota*, and *Campylobacterota*. At the family level of resolution, the most abundant bacterial families are highlighted in Fig. 1C. We note that the overall taxonomic composition of the microbial communities is similar to that observed in other wastewater surveys (1), but the composition of the communities varied appreciably across the sample set. For example, the top three most abundant families across the data set *Comamonadaceae*, *Enterobacteriaceae*, and *Aeromonadaceae* ranged in relative abundances from 5 to 51%, 0.3 to 39%, and 0.07 to 35% per sample, respectively. This high degree of variance in the taxonomic composition of the prokaryotic communities is explored in more detail below. Notably, the viral communities were also highly variable in composition across this sample set with these patterns explored in more detail below.

A relatively small fraction of the prokaryotes identified in these wastewater samples appear to be derived from the human microbiome. This is qualitatively evident by examining the abundances of the bacterial families commonly found in human feces (*Bacteroidaceae*, *Ruminococcaceae*, *Lachnospiraceae*, *Porphyromonadaceae*, *Rikenellaceae*, and *Prevotellaceae* [9]), which collectively account for only 13% of the bacterial and archaeal rRNA gene reads recovered from all sewage samples. To investigate these patterns in more detail, we used the indicator taxon approach described by Barberán et al. (29), which identified particular taxa that are consistently found in human feces and either absent, or present in low abundances, in other sample types, including soil and aquatic environments. The summed abundances of these human microbiome "indicator" taxa represent a relatively small percentage of all 16S rRNA gene reads recovered from the wastewater metagenomes (mean, 8.6% of total reads; range, 0.2 to 35% reads per sample). Thus, whereas bacteria derived from human feces are present in the sewage microbiome, the human microbiome is a reasonably small contributor to the wastewater microbiome, confirming results reported previously (9, 12, 13, 30). Together, these results suggest that a large fraction of the bacteria found in the wastewater samples are more permanent residents of the sewage system and likely derived from biofilms, sediments, or other locations within the sewer system itself. The wastewater microbiome is a unique microbial habitat that harbors distinct microbial communities; it is not exclusively a system transporting transient microbes added into the system from human feces.

We calculated Bray-Curtis dissimilarities in overall functional gene profiles across the sample set from the SqueezeMeta output (31), using normalized gene abundances across 168 unique pathways, and found that overall functional gene composition was well-correlated with the corresponding Bray-Curtis dissimilarities in taxonomic composition (Mantel

$r = 0.82$, $P < 0.001$). Differences in taxonomic composition were closely associated with overall differences in functional gene composition. For this reason, we assume that the observed spatiotemporal patterns in taxonomic composition (described below), mirror observed differences in community-level functional gene composition. However, this is not necessarily true for individual genes or gene categories which could exhibit unique distribution patterns. While there are many individual genes or gene categories worthy of detailed exploration, we focused on one category of genes that are particularly relevant to public health, antibiotic resistance genes, with the spatiotemporal patterns in antibiotic resistance gene profiles described below.

**Spatial patterns in the taxonomic composition of the prokaryotic communities.** The 188 sewage samples included in this study were collected from 17 different locations in the same sewage network across an ~1 km² area (Fig. 1A). Sampling location was a significant predictor of differences in sewage prokaryotic community composition (permutational analysis of variance [PERMANOVA] $R^2 = 0.38$, $P < 0.001$). In particular, we noted 5 general clusters of community types (Fig. 1B) and the taxa driving differences across the 5 clusters are detailed in Fig. S1 in the supplemental material. These location-specific differences in taxonomic composition do not appear to be strongly related to sewer material, sewer depth, or the resident human population served by each sewer since PERMANOVA $R^2$ values were <0.04 for all of these variables. Rather, the spatial variation was most strongly associated with differences in sample pH (Mantel $r = 0.23$, $P < 0.001$) with total suspended solids (TSS) concentrations and sample volume of lesser importance (Mantel $r = 0.13$ and 0.11, respectively, $P = 0.004$ for both variables). The importance of sample pH in structuring the sewage microbial communities is similar to other aquatic and terrestrial environments, where pH has also been demonstrated to be closely associated with shifts in bacterial community composition (32–34). There are clearly other location-specific variables, including unmeasured variables like temperature (13) or specific organic carbon or nutrient concentrations, that could be contributing to the observed spatial patterns. However, it is important to note that the sewage communities found at individual locations are far from static and exhibit pronounced temporal variation, as discussed in more detail below.

We next identified which particular taxa were differentially abundant across the location-specific clusters of community types (Fig. 1B). To do so, we used an indicator taxon approach, focusing on the more abundant taxa that were over-represented in clusters 2 to 5 compared to cluster 1 (which included 11 of 17 sampling locations). These results are presented in Fig. S1. Of note, we observed that samples from the location assigned to cluster 2 had higher relative abundances of *Pseudomonadaceae* than samples from other locations and samples from the three locations assigned to cluster 3 had higher relative abundances of *Arcobacteraceae* and *Selenemonadaceae*. Likewise, samples from the cluster 4 location had higher abundances of *Weeksellaceae* with samples from the cluster 5 location having higher abundances of *Lachnospiraceae* and *Bacteroidaceae* than other locations. Interpreting these observed patterns is difficult since these are relatively broad taxonomic groups with diverse physiologies. However, these patterns do suggest that there are location-specific signatures in the sewage bacterial communities, signatures that are most likely a product of the unique biogeochemical conditions found at these sampling locations.

**Temporal patterns in the taxonomic composition of the prokaryotic communities.** The communities found in sewage samples collected from a given location were highly variable in taxonomic composition over the 3-week sampling period. This is evident from Fig. 2, left which highlights that the average Bray-Curtis dissimilarity in taxonomic composition within a given sampling location over time (mean = 0.39) was not appreciably lower than the average Bray-Curtis dissimilarity in taxonomic composition between sampling locations (mean = 0.47). In other words, despite observing significant differences in community composition across sampling locations (as detailed above), the variation observed within a given location over time was similar in magnitude to the differences between sites, with some sampling locations (e.g., sites R and O) exhibiting higher variance in taxonomic community composition than others (e.g.,

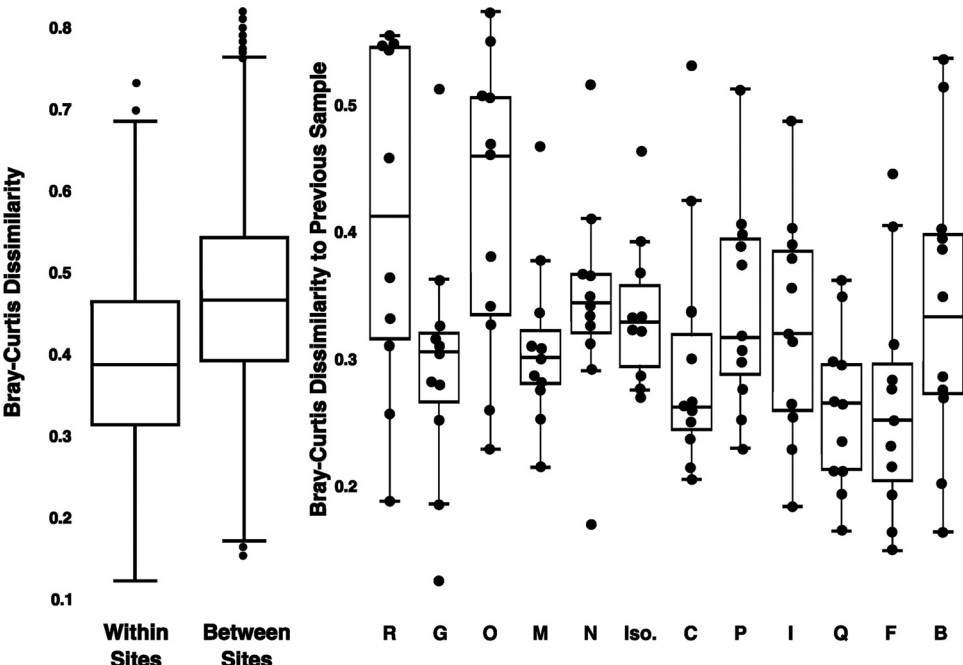

**FIG 2** Temporal variation in overall community composition within individual sampling locations. We summarized pairwise Bray-Curtis distances in the taxonomic composition of the prokaryotic communities within sites (temporal variation) versus between sites (spatial variation). Also shown are the pairwise Bray-Curtis distances in community composition across sampling dates within each individual sampling location. Note that some of the sampling locations (e.g., sites O and R) show higher temporal variance in taxonomic community composition than others (e.g., sites C and F).

sites F and C) (Fig. 2, right). Perhaps more importantly, the temporal patterns observed in the taxonomic composition of the wastewater communities were not consistent across sites. Sampling dates when we observed pronounced changes in the composition of the wastewater communities (compared to the average composition at a given location) were rarely shared across sites (see Fig. S2). We next sought to determine which variables (days since first collection, daily high/low temperatures, the volume of sewage collected, sewage pH, daily precipitation amounts, or total suspended solid concentrations) best explained the temporal variation observed at individual sampling locations. Using Mantel tests, "days since" was significant at 9 of the 15 locations and TSS was significant at 6 of the 15 locations ($R > 0.25$ and $P < 0.05$) (see Table S1). No other variable was significant at more than two individual sites. Thus, Mantel tests indicate that "days since" and TSS are the factors most strongly correlated with temporal variation in community composition. These results are consistent with the results from the MRM models which were variable in their overall explanatory power ($R^2$ values from 0.09 to 0.74) with the four most significant scores ($R^2 \geq 0.25$, $P < 0.01$) each indicating one or both of "days since" and TSS in the top two variables (see Table S1). Although temperature has been noted previously as being important in structuring wastewater communities (13), changes in daily high and low temperatures were not strongly associated with observed changes in community composition across any of the sites. We note, however, that surface air temperature may not reflect the temperature of the wastewater at these upstream locations.

The associations between temporal changes in community composition and both days from the start of the study and total suspended solid concentrations are most likely driven by differences in usage patterns, most notably temporal changes in human fecal inputs into the sewage system. This is evident from an increase in the relative abundance of human fecal indicator bacteria in week 2 compared to weeks 1 and 3, a pattern that was consistent across most sites (see Fig. S3). We hypothesize that this week 2 increase in human fecal indicator bacteria is a product of the large early-season

snow event that occurred 8 and 9 September 2020, which may have increased the ratio of blackwater to graywater by altering residential bathing habits through a notable temperature decrease from 33 to 0°C within 18 h. Together, these results highlight that there is pronounced temporal variation observed at individual sampling locations, and this variation is most likely associated with changes in fecal inputs to the system (although fecal bacteria represent a relatively small proportion of taxa in these communities [see discussion above]), reflecting the variation noted in other studies on short temporal timescales (12). We acknowledge that most of the observed temporal variation at the individual sampling locations remains statistically unexplained, highlighting the importance of collecting additional information on wastewater systems and their usage. There are likely other unmeasured factors, including sewer temperatures, sewage inputs, or specific usage patterns, that could be used to predict when and how wastewater communities will change over time (13). More generally, these results emphasize the importance of considering temporal variation when assessing spatial differences in wastewater communities, even in a given sewer network. A single time point sampling will give a limited, and perhaps even misleading, perspective on the composition of wastewater prokaryotic communities.

An additional factor that may have contributed to the observed temporal variability is the fact the communities may not have reached a steady-state for this university sewer network. Samples were collected within 1 to 2 weeks after students returned to on-campus residences, following a nearly stagnant and unused sewer system from March to August 2020. This flushing of stagnant water over time may explain why "days since the start of sampling" was often a factor associated with the observed temporal variation in bacterial community composition (see Table S1), a result consistent with our hypothesis that flushing of stagnant water, lack of residual disinfectant, and idle plumbing can contribute to changes in wastewater microbiome composition (35). More specifically, we expected that flushing of stagnant water could be associated with increases in the abundances of premise-plumbing associated pathogens such as *Legionella pneumophila* and *Acinetobacter baumannii* (36–38). For the first 5 days of the sampling period, when stagnant water was likely being flushed through the system, we found that there was an initial spike followed by a consistent decrease in the relative abundances of *Legionella* at 10 of the 15 sampling locations and of *Acinetobacter* at 7 of 15 sample locations (see Fig. S4). However, additional work is needed to investigate these dynamics in more detail and to determine the public health relevance of these or other potential pathogens accumulating in plumbing or sewer systems upon prolonged stagnation.

**Spatiotemporal patterns in the abundances of antibiotic resistance genes.** Given the public health importance of documenting antibiotic resistance genes (ARGs) in wastewater and other built environments (8, 39, 40), we focused our functional gene analyses of the metagenomic data on those genes presumed to be associated with antibiotic resistance. However, we note that we are likely missing other potentially important ARGs and not all genes annotated as ARGs necessarily confer antibiotic resistance (41). The number of metagenomic sequences classified as ARGs was low, with ARGs representing an average of 0.003% of reads per sample (range, 0.0003 to 0.010%) (Fig. 3A). This low fraction of ARGs detected is similar to what has been observed in comparable metagenomic data sets (16, 42, 43). Even with these low abundances, we detected a broad diversity of ARGs across the sample set, including ARGs associated with resistance to 27 different drug classes, but the proportional abundances of ARG types were reasonably consistent across the locations sampled (Fig. 3B). The most abundant ARGs detected are those associated with resistance to tetracycline, macrolide, and beta-lactam drug classes (Fig. 3B). These dominant classes of ARGs were also recovered in metagenomic analyses of similar sample types (16), but we note that this may be, in part, a product of the reference database being biased toward ARGs that are well characterized.

We did observe pronounced variation in the relative abundance of ARG genes

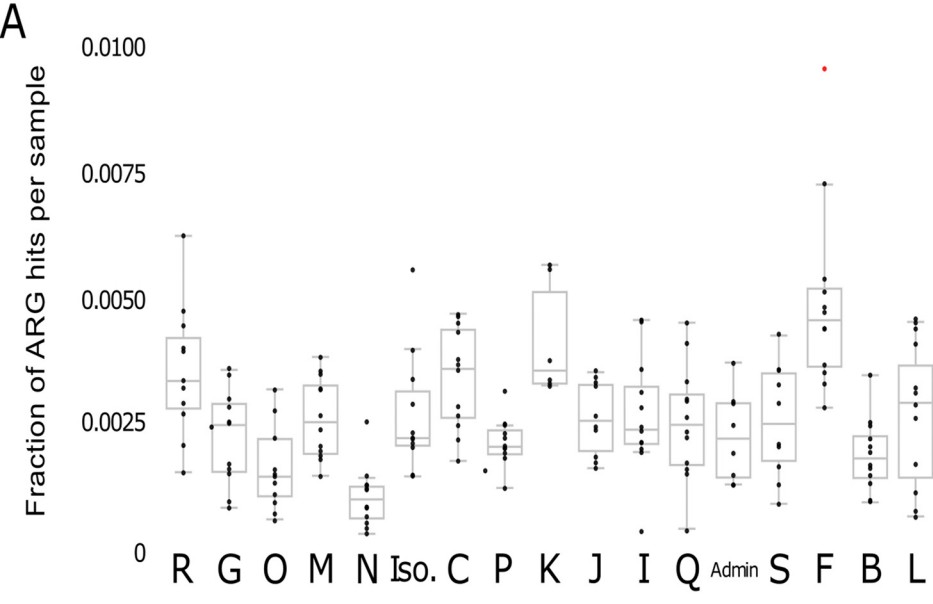
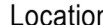
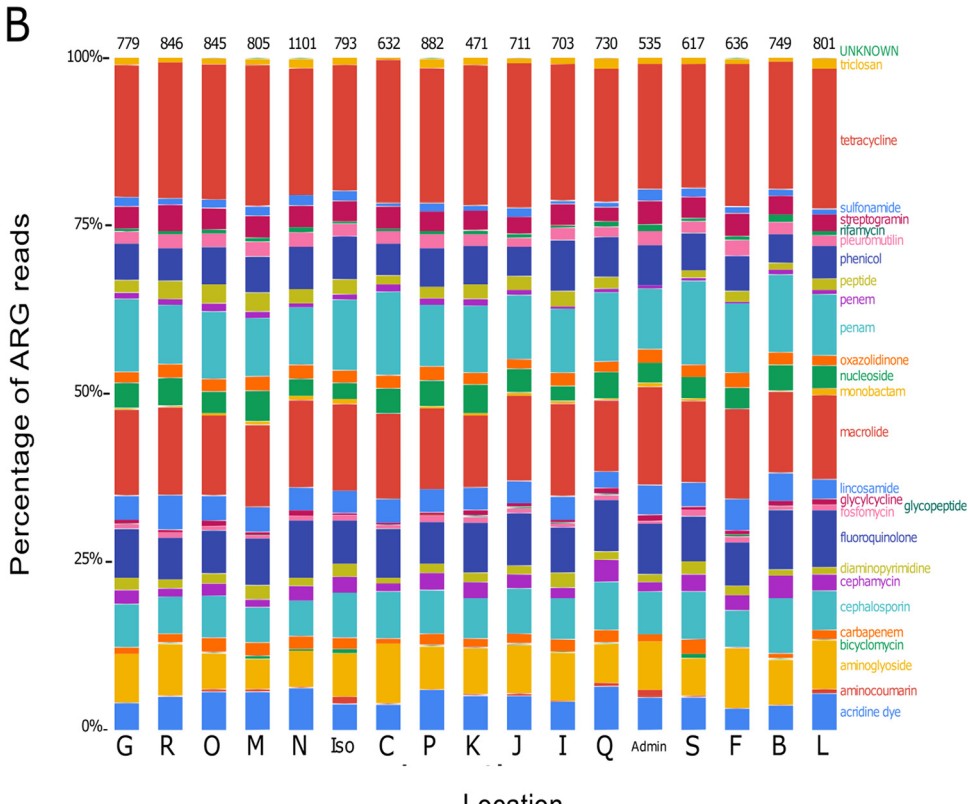

**FIG 3** (A) Variation in the proportion of reads assigned to antibiotic resistance genes (ARGs) normalized to total read count at each sampling location. We observed a significant effect of sampling location on ARG abundances (one-way ANOVA, F value = 8.06, $P < 0.001$). (B) Proportional abundances of ARGs assigned to different drug classes, highlighting the consistency in general ARG types across locations.

across sampling locations and over time within individual locations (Fig. 3A). Given that ARGs are often considered to be particularly abundant in human fecal material-derived bacteria, we hypothesized that those samples with higher abundances of fecal bacteria would have higher relative abundances of ARGs. This was not the case, as the

correlation between ARGs and fecal indicator abundances across the whole sample set was relatively weak (linear regression, $R^2 = 0.14$, $P < 0.001$). While ARGs are clearly present in wastewater samples, these ARGs are not necessarily indicative of fecal contamination, and the ARGs are often likely to be associated with other bacteria residing in the sewer system. Not all ARGs are necessarily derived from anthropogenic inputs and presumably ARGs would be present in this system even in the absence of appreciable fecal inputs (as evidenced from the abundance of ARGs detected at the "Admin" site which served a building with few if any occupants during the sampling period). More work is needed to identify which bacteria in sewage systems harbor particular ARGs, where these bacteria reside within sewer systems, and the public health relevance of these sewer-associated ARGs.

**Spatiotemporal patterns in viral communities.** In addition to investigating the composition of the prokaryotic communities and how they vary across sampling locations or within individual locations over the 3-week period, we also characterized the corresponding spatiotemporal variation in the viral communities. This study was motivated by the likely importance of bacteriophage to the structure of sewage and other microbiomes (15, 44). Moreover, with the growing interest in documenting viral community dynamics in built environments (45), it is important to assess how spatiotemporal variation in viral community structure compares to the corresponding variation in prokaryotic communities which are far more commonly studied. Viral communities were characterized using FastVirome Explorer with the IMG/VR and NCBI RefSeq databases (see Materials and Methods). We start by noting three important caveats associated with the viral analyses. First, as our analyses were restricted to double-stranded DNA (dsDNA), we did not capture RNA or single-stranded DNA (ssDNA) viruses. Second, we are likely identifying only the more abundant viruses in these samples since a deeper sequencing depth would be required to identify relatively rare viral taxa in these samples. Third, since we used a reference viral genome database for classifying viral reads, our analyses are restricted only to viruses included in the reference database, and we would not capture novel viral diversity, which is likely high in these types of environments (46). Despite these caveats, we were able to document pronounced spatiotemporal variation in the viral communities.

The dominant viral taxa identified when using the IMG/VR database were within the order *Caudovirales* (tailed dsDNA bacteriophages), representing 99.40% of the viral reads that could be classified. While 68.63% of *Caudovirales* reads were not resolved to family level, three families within this order—*Myoviridae*, *Podoviridae*, and *Siphoviridae*—represented 12.00, 10.86, and 7.49% of total viral reads, respectively. These viral taxa have also been found to be dominant in another metagenomic-based sewage survey (15). Using results from analyses conducted with the NCBI RefSeq database, we recovered sequences assigned to the crAssphage (cross-assembly phage) group (69% of viral reads on average; see Fig. S5A), which is notable given that this viral group is often considered an indicator of fecal contamination (47). Importantly, we observed variation in the proportional abundances of the dominant viral taxa recovered across the sampling locations as annotated with both the NCBI RefSeq database (see Fig. S5A) and the IMG/VR database (see Fig. S5B).

The observed variation in the composition of the viral communities was reasonably well-correlated with the variation in prokaryotic community composition. This was evident when we compared viral to prokaryotic community composition across all samples (Mantel $r = 0.67$, $P = 0.001$) and within individual locations over time (Mantel $r = 0.42$ to $0.87$, $P < 0.001$ in all cases). This correspondence between viral and prokaryotic community composition is not surprising given the host specificity of many bacteriophage (48, 49) with similarly close associations between sewage phage and prokaryotic community composition having been observed previously (15). Moreover, the close correspondence between viral and prokaryotic community composition indicates that the spatiotemporal patterns in prokaryotic communities (as described above) apply to both components of the sewage microbiome with viral communities also exhibiting pronounced variation in composition across sampling locations and over time within individual locations.

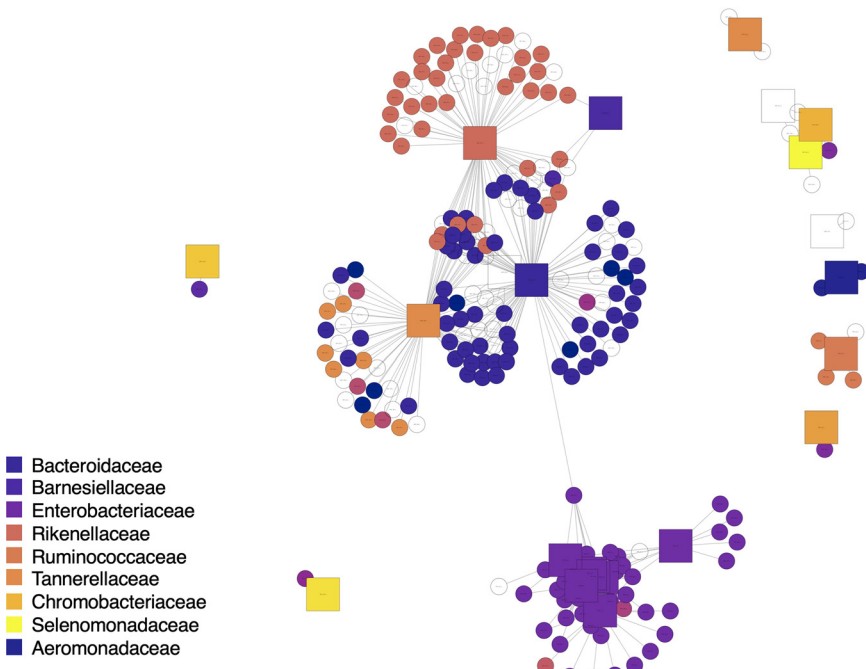

**Bacteroidaceae**
**Barnesiellaceae**
**Enterobacteriaceae**
**Rikenellaceae**
**Ruminococcaceae**
**Tannerellaceae**
**Chromobacteriaceae**
**Selenomonadaceae**
**Aeromonadaceae**

**FIG 4** Bipartite cooccurrence network of top viral and bacterial taxa detected across all sewage samples (only viral and bacterial taxa correlated with a Pearson r value > 0.75 are included here). Square nodes are bacterial taxa colored by bacterial family. Circular nodes are viral (phage) nodes colored by putative bacterial host as assigned in the IMG database, illustrating the accuracy of phage host assignments.

To investigate the associations between viral and prokaryotic communities in more detail, we used network analyses based on cooccurrence patterns to identify viruses (including those with both known and undetermined bacterial hosts) that consistently cooccur with particular prokaryotic taxa. Although cooccurrence patterns provide only putative associations between particular viruses and their potential bacterial hosts, we were able to accurately reconstruct known phage-host relationships (Fig. 4). For example, we identified a number of phage taxa known to infect bacteria within the *Enterobacteriaceae* and *Bacteroidaceae* families (Fig. 4) that also cooccurred with these bacterial families across the sample set. However, there were a few viruses that cooccurred with multiple distinct bacterial taxa, a result we might expect given that some phage can infect a broad range of bacterial hosts (48, 49). Likewise, some of the viruses with undetermined host specificity cooccurred with specific bacterial taxa (Fig. 4), highlighting that these cooccurrence analyses can be used to generate hypotheses about the likely host preferences of poorly characterized viral taxa.

**Conclusions.** By conducting shotgun metagenomic analyses on 188 wastewater samples collected over a 3-week period from 17 locations within a single sewage network on a university campus, we were able to document the spatial and temporal patterns in prokaryotic and viral communities at reasonably high spatiotemporal resolution. As overall differences in functional gene profiles closely mirrored differences in the taxonomic composition of the prokaryotic communities (see "General Description of the Prokaryotic Communities" above), we focused our functional gene analyses on genes putatively coding for antibiotic resistance functions given their potential importance. Together, the results from these analyses lead us to several broader conclusions. First, sewage microbial communities are not dominated by taxa derived from human fecal inputs. Rather, many of the taxa found in sewage are likely derived from sewer biofilms or accumulated sediments within the sewer system, highlighting the importance of understanding how these resident microbial communities develop over time and across different locations within an individual sewer system. Second, we observed pronounced spatial variation, even across locations within the same network in close proximity,

but the spatial variation was often similar in magnitude to the temporal variation observed at a given sampling location, a pattern that was observed for prokaryotic and viral community composition as well as ARG profiles. Just as importantly, the temporal patterns in the sewage communities were often inconsistent across sampling locations with different patterns (and different abiotic variables associated with those patterns) across the sampling locations. This high degree of temporal variation, variation that is often difficult to predict *a priori*, is relevant to future work since it highlights the importance of collecting additional information to characterize the sewage environment across space and time, including specific usage patterns in buildings served by a given sewer system (e.g., frequency of toilet or appliance usage), sewage nutrient or xenobiotic concentrations, temperature, and other variables that were not directly measured for this study. Presumably, additional information describing conditions in the sewage system, or the buildings feeding the sewage system, would provide more insight into the specific factors associated with the observed variation in sewage microbial communities. More generally, these results highlight that collecting sewage samples from a single time point, or just a few time points, may not adequately capture microbial dynamics at a given location, and studies that do not explicitly characterize potential temporal variation when assessing spatial variation may yield an incomplete perspective on the sewage microbiome.

## MATERIALS AND METHODS

**Sample collection.** Wastewater samples were collected from 17 locations across the University of Colorado campus in Boulder, CO (40.00°N, 105.26°W). All locations are within the same sewer network with 14 of the 17 sampling locations representing sewage outflow from individual buildings with the remaining locations representing flow from multiple buildings, meaning they were either downstream of several buildings in the sewer network, or regularly received backflow from several buildings (Fig. 1A). Sampling was conducted from surface-accessible manholes with wastewater collected at each location using composite autosamplers (50). Sampling locations are labeled followed the naming convention used previously (50). All samples included in this study were collected over a 3-week period (1 to 20 September 2020) and up to four samples were collected per week (Tuesday, Thursday, Saturday, and Sunday) for a total of 6 to 12 sampling dates per location (13 of the 17 locations had at least 11 sampling time points). Each sample represented a 24-h flow-proportional composite sample (50). In total, we collected 188 sewage samples, with each sample representing an integrated, continuous collection over a 24-h period withdrawn using peristaltic pumps. The total volume of sewage collected from each location on each sampling date (mean, 8 L; range, 0.3 to 16.2 L) represented a proxy for the total volume of sewage moving through each location over each 24-h period, withdrawing lower volumes when the sewer pipe was dry or sampling inlet mispositioned. All samples were stored on ice during the 24 h withdrawal and immediately upon collection. After transportation to the laboratory, 2-mL aliquots were withdrawn and centrifuged at 14,000 $\times$ $g$ for 10 min at 4°C, and the decanted pellets with were frozen at $-80$°C within 4 h of collection for subsequent DNA analyses.

In addition to the shotgun metagenomic analyses described below, we also measured the pH of each sample (mean pH, 8.4; range, 7.0 to 9.6) and the amount of total suspended solids (mean, 80 mg L$^{-1}$; range, 10 to 300 mg L$^{-1}$) using standard methods (50). Additional data collected for each sampling location included: sewer material (brick versus concrete), sewer depth (the depth below the ground surface from which samples were collected [mean, 3.0 m; range, 1.5 to 5.5 m]), and the estimated population of full-time residents served by that particular sewer line (mean, 544 individuals; range, 0 to 2,536 individuals; with only one location, a classroom/office building, having no full-time residents). For the temporal analyses, we also included information on daily high/low air temperatures and precipitation amounts with these data compiled from the National Oceanic and Atmospheric Administration Physical Sciences Laboratory.

**Shotgun metagenomic sequencing.** DNA was extracted from homogenized 2-mL aliquots of each of the 188 samples with two DNA extraction "blanks" and two no-template control samples processed alongside the samples to check for potential contamination. DNA was extracted using a DNeasy PowerSoil HTP 96 kit (Qiagen, Germantown, MD) according to the manufacturer's instructions, and libraries for shotgun metagenomic sequencing were prepared using Illumina DNA Prep kits and IDT for Illumina DNA Unique Dual Index Sets (Illumina, San Diego, CA). Briefly, DNA was fragmented and adapter tag sequences were added using bead-linked transposomes, followed by a PCR step to add adapter sequences. The resulting libraries were then cleaned by bead purification, quantified, and pooled in equimolar concentrations. Pooled libraries (which included 188 sewage samples and the 4 "blank" samples) were sequenced on duplicate Illumina NovaSeq lanes running the 2 $\times$ 150-bp paired-end sequencing chemistry. This sequencing effort yielded 3.4 to 24 million paired-end reads per sewage sample (mean, 10 million) with the four "blank" samples yielding 0 to 1,700 reads, indicating nonexistent to minimal contamination.

**Bioinformatics.** Read quality was assessed using FastQC (51) and adapters were removed using Cutadapt version 2.4 (52). We used Sickle (53) to trim bases of insufficient quality (q-score $>$20) and reads of insufficient length ($<$50 bp). After these quality control and filtering steps, a total of 983 million

paired-end reads remained across all 188 sewage samples, with a mean of 5.23 million reads per sample (range, 1.66 to 10.64 million).

To assess the taxonomic composition of the communities, we used phyloFlash (28), a tool that extracts small-subunit rRNA gene reads from the shotgun metagenomic data, with taxonomic classification of the reads determined by comparing extracted reads against those in the SILVA reference database, version 138.1 (54). As nearly all of these reads were prokaryotic (bacterial and archaeal), with eukaryotic reads representing only 0.26% of all rRNA gene reads across the data set, we removed reads from the resulting taxon table that were classified as chloroplasts, mitochondria, or eukaryotes (this process removed only 18,475 of the 2,156,483 rRNA gene reads across the entire data set). We also removed those taxa from the table that were represented by fewer than 10 reads across the entire data set to be conservative in our assessment of prokaryotic diversity (removing an additional 4,469 reads in total). The resulting taxon table yielded a mean of 11,370 prokaryotic rRNA gene reads per sample (range, 1,702 to 22,056). To provide a broad assessment of how functional gene profiles may relate to the taxonomic composition of the communities, we annotated functional gene reads with the SqueezeMeta pipeline's "read-only" analysis script, sqm_reads.pl (31), to obtain KEGG annotations of genes. To determine the presence of unique genes, we then used the KEGG ontology (KO) with gene abundances normalized against the abundances of single-copy genes with MUSiCC (55). We identified >17,000 unique genes (KOs) across the entire data set. These KOs collectively represented 168 unique gene pathways with Bray-Curtis dissimilarities calculated from the assigned normalized abundances of genes per sample.

We quantified the classes and relative abundances of antibiotic resistance genes in each sample by processing trimmed and filtered metagenomic reads through the Resistance Gene Identifier pipeline (56) using default parameters for shotgun metagenomic data and the Comprehensive Antibiotic Resistance Database (CARD; version 3.0.1) as the reference database. After assigning ARGs, we removed "rare" genes (those that occurred in <10 of the 188 samples) and those deemed to be of insufficient quality (<40 MAPQ score).

For the viral community analyses, we used FastViromeExplorer (57) with quality-filtered reads against the IMG/VR database (58) and, separately, the NCBI RefSeq database to detect and identify viruses (dsDNA phages) present in the sewage metagenome samples. We used these two different databases to characterize as many viral reads as possible due to database differences in annotation and taxonomic resolution. Specifically, 1,921,070 of our reads matched against the NCBI database, while 8,961,471 of our reads matched against the IMG/VR database. However, the taxonomic resolution and species identification varied between the databases. For all analyses, with the exception of the calculations of crAssphage abundances, we used the IMG/VR annotation due to the larger amount of reads characterized. As the number of recovered viral reads per sample was correlated with overall per-sample sequencing depth ($r^2 = 0.72$, $P < 0.001$), we rarefied to 2206 viral reads per sample and removed viral taxa represented by <10 reads across the entire data set prior to downstream analyses. As explained in more detail above (see Results and Discussion), we note that this pipeline would be expected to underestimate the total amount of viral diversity that could be found in these samples. A bipartite cooccurrence network of the more abundant viral and bacterial taxa was constructed in R using the iGraph package ("graph_from_incidence_matrix") (59). Input for the bipartite network was a correlation table of viral-bacterial abundances across samples (filtered to include only correlations with Pearson $r$ values > 0.75).

**Data visualization and statistical analyses.** All downstream analyses were conducted in R (60), unless otherwise noted. For the spatial analyses, we calculated average pairwise Bray-Curtis dissimilarity levels across the 17 sampling locations, with differences across sample categories determined using PERMANOVA as implemented in the R package vegan 2.5.7 (61). Location-specific differences in community composition were visualized via hierarchical clustering (Ward method, R package Stats 4.0.5). To determine what measured sewage variables (including sampling date, weekend versus weekday, location, pH, sewer depth, total suspended solids, sewer age, sewer material, resident population, sample volume, geographic location, and total read counts) were associated with the observed variation in the taxonomic composition of bacterial communities, we used Mantel tests (R package vegan 2.5.7) with Euclidean distances in the measured continuous variables and Bray-Curtis distances in the taxonomic composition of the microbial communities. Analysis of volatility in taxonomic composition was performed in Python 3.6.9, with Bray-Curtis distances calculated via the package scikit-bio version 0.5.6. Analyses of the temporal variation in community composition were restricted to samples from 15 of the 17 sites as these 15 sites had samples from at least 10 of the 12 total sampling times over the 3-week period (2 sites—"K" and "Admin"—were represented by only 9 and 6 samples, respectively). We used both Mantel tests (R package vegan v2.5.7) and multiple regression on distance matrices (MRM, R package ecodist v2.0.7) to identify the specific variables that explain the observed temporal variation in bacterial community composition within sites.

**Data availability.** Raw sequence reads for all samples and the associated sample metadata have been uploaded to the NCBI Sequence Read Archive repository under accession number PRJNA875025.

## SUPPLEMENTAL MATERIAL

Supplemental material is available online only.
**FIG S1**, PDF file, 0.7 MB.
**FIG S2**, PDF file, 0.03 MB.
**FIG S3**, PDF file, 0.1 MB.
**FIG S4**, PDF file, 0.03 MB.
**FIG S5**, PDF file, 0.2 MB.
**TABLE S1**, PDF file, 0.04 MB.

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
