## [Reviewer comments · mSystems]

A metagenomic investigation of spatial and temporal changes in sewage microbiomes across a university campus

Noah Fierer, Hannah Holland-Moritz, Alexandra Alexiev, Harpreet Batther, Nicholas Dragone, Liam Friar, Matthew Gebert, Sarah Gering, Jessica Henley, Sierra Jech, Emily Kibby, Tina Melie, William Patterson, Eric Peterson, Kyle Schutz, Elias Stallard-Olivera, John Sterrett, Corinne Walsh, and Cresten Mansfeldt

Corresponding Author(s): Noah Fierer, University of Colorado Boulder

Review Timeline:

Submission Date:	July 12, 2022
Editorial Decision:	August 22, 2022
Revision Received:	August 30, 2022
Accepted:	September 1, 2022

Editor: Rachel Poretsky

Reviewer(s): The reviewers have opted to remain anonymous.

Transaction Report:

DOI: <https://doi.org/10.1128/msystems.00651-22>

August 22, 2022

Dr. Noah Fierer
University of Colorado Boulder
Ecology and Evolutionary Biology
Boulder, CO

Re: mSystems00651-22 (A metagenomic investigation of spatial and temporal changes in sewage microbiomes across a university campus)

Dear Dr. Noah Fierer:

Thank you for submitting your manuscript to mSystems. We have completed our review and I am pleased to inform you that, in principle, we expect to accept it for publication in mSystems. However, acceptance will not be final until you have adequately addressed the reviewer comments.

Preparing Revision Guidelines

Sincerely,

Rachel Poretzky

Editor, mSystems

Journals Department
American Society for Microbiology
1752 N St., NW

Reviewer comments:

Reviewer #1 (Comments for the Author):

Overall, I find that the manuscript is nicely written and easy to follow. Although many of the results did not lead to novel conclusions about sewage microbial and gene patterns, they did confirm several patterns described previously, which is important as it suggests these patterns are robust. Also, the temporal and spatial scales examined in this manuscript are unique in that they combine a short time-series with a multi-location sampling campaign in a single sewer network. To my knowledge this type of sampling regime has not been common in other sewage microbial community papers, and thus it adds to a growing body of knowledge about community and gene variability in sewage conveyance systems.

I do not have major comments for the authors to address but have some more minor questions for their consideration.

P4 Shotgun metagenomic sequencing - The methods do not describe or cite what method was used for fragmenting the DNA in the library preparation step or what was used to clean the DNA after library preparation. These should be described.

P7 - On this page there is mention of using an indicator taxon approach for classifying taxa that are consistently found in human feces to understand the human fecal content in each sample. The results presented are in line with what others have found, but this method is not described in the methods section. Here it is listed that an indicator taxon approach was followed as described in reference 42. A parameter setting and R package for the indicator analysis is listed in that paper, but in part these analyses depend on the sequence database used for the source environment. Tracing back a few paper references, it appears the human database is from Costello et al., 2009 and this may have been combined with other animal fecal datasets in the ref 42 paper. Given this chain of papers and slightly changing methods across the papers, the authors should add more specifically what was used for the human fecal database and indicator taxon analysis to the methods this paper.

P8 - On this page, there is mention that sample volume is a small but significant factor in determining bacterial community composition. Given that the sample volumes ranged from 0.3 L to 16.2 L, I wonder if sample volume had any impact on community variability. The data suggests some community cohesion at individual sites (i.e. lower within site variability), but it was not that strong. So, does within-site sample volume variability lead to increased microbial community variability? Or in another phrasing, do sites with low sample volume variability have lower community variability over time? I ask because I wonder about the intersection of changing flow dynamics at a site and its influence on community dynamics versus additional variability added by sampling (which happens because it is not simple to sample these locations under variable flow). The manuscript states that flow was not related to community composition, but I wonder if it would relate more to changes in the community at a single site. Additional thoughts by the authors on this topic would be appreciated, since I believe at these upstream sites sample collection differences could have an impact on the resultant community composition.

P9 - P10 top - Here the manuscript states that there was stagnant water at the beginning of the sampling period and thus communities may not have reached steady-state in this network over the sampling period. The *Legionella*/*Acinetobacter* data suggest this is possible. If this point is correct, then this section argues against the main conclusion that temporal variability impacts spatial sewage analysis studies and should be accounted for. I suggest the authors clarify their thinking here, which would also tie this section into the rest of the results/discussion. Right now, the paragraph seems to argue that the unusual circumstance of long-term stagnant water was the factor driving temporal variability in the samples, but this idea is not mentioned later in the conclusions. It may be possible to examine whether community variability decreased over time - if so, then maybe this stagnant water had a large impact.

Figure 1 - this is not important for the review but note the image quality on Figure 1A makes it difficult to decipher the letters indicating each sampling location.

Throughout manuscript - italics is needed for valid taxonomic names. Although there is not consistent implementation by authors or journals for that matter, bacterial/archaeal family names should be italicized in the text. The convention for bacteria/archaea is to differentiate all ranks of taxonomy when printed in text (see Bergey's Manual or Oren & Garrity 2021. *Int. J. Syst. Evol. Microbiol.* Again, I know this is not always followed, but it would make me happy to have italics for taxonomic names be more consistent across papers.

Reviewer #2 (Comments for the Author):

General remarks:

This study reveals some interesting observations in spatiotemporal variations of sewage microbial communities. However, it can

benefit from more in-depth discussion in terms of factors contribute to the dynamics and possible mechanisms.

Specific comments:

Section 3.1, paragraph 2: "Together these results suggest that a large fraction of the bacteria found in the wastewater samples are more permanent residents of the sewage system and likely derived from biofilms, sediments, or other locations within the sewer system itself." Please provide more evidence to support this argument.

Section 3.2, paragraph 1: "Rather, the spatial variation was most strongly associated with differences in sample pH (Mantel $r = 0.23$, $P < 0.001$) with total suspended solids (TSS) concentrations and sample volume of lesser importance (Mantel $r = 0.13$ and 0.11 , respectively, $P = 0.004$ for both variables)." This is interesting observation. What are the major causes of pH and TSS variation in these sewage sampling points? It is worth discussing these leading factors for the differences in sewage microbial communities.

Section 3.2, paragraph 2: It is not clear to me how these 5 clusters were defined, e.g., what is the cut-off or criteria? In addition, do the site labels have meanings? Since there are clusters, I am curious if there are some common characteristics or patterns can be concluded? For example, do the residential buildings or teaching/lab buildings tend to cluster together? The discussion regarding the spatial variation is not sufficient in my opinion.

Section 3.3, paragraph 3: The observation of initial spikes of *Legionella* and *Acinetobacter* is very interesting. Would this be a health concern for the occupants? Was this also observed from other studies? If so, should an initial flushing of the stagnant systems be recommended for places like dormitories?

Other comments:

Please add line numbers for easy of reviewing and commenting.

Figures are in general very blurry. Please fix this in future submission.

We thank the reviewers for their time and energy. Their comments were thoughtful and comprehensive. We have responded to all of the reviewer comments (in bold, italicized text below) and have revised the manuscript accordingly.

Reviewer #1 (Comments for the Author):

Overall, I find that the manuscript is nicely written and easy to follow. Although many of the results did not lead to novel conclusions about sewage microbial and gene patterns, they did confirm several patterns described previously, which is important as it suggests these patterns are robust. Also, the temporal and spatial scales examined in this manuscript are unique in that they combine a short time-series with a multi-location sampling campaign in a single sewer network. To my knowledge this type of sampling regime has not been common in other sewage microbial community papers, and thus it adds to a growing body of knowledge about community and gene variability in sewage conveyance systems.

I do not have major comments for the authors to address but have some more minor questions for their consideration.

We appreciate the positive comments.

P4 Shotgun metagenomic sequencing - The methods do not describe or cite what method was used for fragmenting the DNA in the library preparation step or what was used to clean the DNA after library preparation. These should be described.

We have now added a more detailed explanation of the library prep method as we agree that these details are important.

P7 - On this page there is mention of using an indicator taxon approach for classifying taxa that are consistently found in human feces to understand the human fecal content in each sample. The results presented are in line with what others have found, but this method is not described in the methods section. Here it is listed that an indicator taxon approach was followed as described in reference 42. A parameter setting and R package for the indicator analysis is listed in that paper, but in part these analyses depend on the sequence database used for the source environment. Tracing back a few paper references, it appears the human database is from Costello et al., 2009 and this may have been combined with other animal fecal datasets in the ref 42 paper. Given this chain of papers and slightly changing methods across the papers, the authors should add more specifically what was used for the human fecal database and indicator taxon analysis to the methods this paper.

We have decided to keep the reference to the Barberán et al. paper as it best describes the indicator taxon approach and the reference databases used to identify fecal-associated bacteria. The reviewer is correct that the human fecal data was from Costello et al. 2009, but the key to using this approach is to identify taxa that are common and nearly ubiquitous in fecal samples, but not found in soil or aquatic environments (which have their own respective databases). Thus, we think it is most appropriate to cite the Barberán et al. paper as it provides full details on the databases used and how indicator taxa were identified.

P8 - On this page, there is mention that sample volume is a small but significant factor in determining bacterial community composition. Given that the sample volumes ranged from 0.3 L to 16.2 L, I wonder if sample volume had any impact on community variability. The data

suggests some community cohesion at individual sites (i.e. lower within site variability), but it was not that strong. So, does within-site sample volume variability lead to increased microbial community variability? Or in another phrasing, do sites with low sample volume variability have lower community variability over time? I ask because I wonder about the intersection of changing flow dynamics at a site and its influence on community dynamics versus additional variability added by sampling (which happens because it is not simple to sample these locations under variable flow). The manuscript states that flow was not related to community composition, but I wonder if it would relate more to changes in the community at a single site. Additional thoughts by the authors on this topic would be appreciated, since I believe at these upstream sites sample collection differences could have an impact on the resultant community composition.

Great question. The temporal variability in community composition observed at individual sites was never significantly correlated with measured variation in flow rates (which was relatively consistent within sites over time). We have now added text to highlight this point.

P9 - P10 top - Here the manuscript states that there was stagnant water at the beginning of the sampling period and thus communities may not have reached steady-state in this network over the sampling period. The Legionella/Acinetobacter data suggest this is possible. If this point is correct, then this section argues against the main conclusion that temporal variability impacts spatial sewage analysis studies and should be accounted for. I suggest the authors clarify their thinking here, which would also tie this section into the rest of the results/discussion. Right now, the paragraph seems to argue that the unusual circumstance of long-term stagnant water was the factor driving temporal variability in the samples, but this idea is not mentioned later in the conclusions. It may be possible to examine whether community variability decreased over time - if so, then maybe this stagnant water had a large impact.

Another great point. We have added text as recommended to that paragraph to explain how flushing of stagnant water may have contributed to the observed temporal patterns.

Figure 1 - this is not important for the review but note the image quality on Figure 1A makes it difficult to decipher the letters indicating each sampling location.

We have uploaded higher quality versions of all figures.

Throughout manuscript - italics is needed for valid taxonomic names. Although there is not consistent implementation by authors or journals for that matter, bacterial/archaeal family names should be italicized in the text. The convention for bacteria/archaea is to differentiate all ranks of taxonomy when printed in text (see Bergey's Manual or Oren & Garrity 2021. Int. J. Syst. Evol. Microbiol. Again, I know this is not always followed, but it would make me happy to have italics for taxonomic names be more consistent across papers.

Changes made as recommended.

Reviewer #2 (Comments for the Author):

General remarks:

This study reveals some interesting observations in spatiotemporal variations of sewage microbial communities. However, it can benefit from more in-depth discussion in terms of factors contribute to the dynamics and possible mechanisms.

Specific comments:

Section 3.1, paragraph 2: "Together these results suggest that a large fraction of the bacteria found in the wastewater samples are more permanent residents of the sewage system and likely derived from biofilms, sediments, or other locations within the sewer system itself." Please provide more evidence to support this argument.

We show that the dominant taxa found in human feces were relatively rare in the wastewater samples. In addition, we used a quantitative, indicator taxon approach to directly compare the taxa identified in the sewage communities to those taxa that are common in human feces, but not found in other sample types. Both analyses confirm that human-derived bacteria were relatively rare in the wastewater samples and thus the bacteria are most likely derived from other sources. This information is included earlier in the paragraph cited by the reviewer.

Section 3.2, paragraph 1: "Rather, the spatial variation was most strongly associated with differences in sample pH (Mantel $r = 0.23$, $P < 0.001$) with total suspended solids (TSS) concentrations and sample volume of lesser importance (Mantel $r = 0.13$ and 0.11 , respectively, $P = 0.004$ for both variables)." This is interesting observation. What are the major causes of pH and TSS variation in these sewage sampling points? It is worth discussing these leading factors for the differences in sewage microbial communities.

Great question. TSS variation is most likely associated with variation in flow rates and inputs into each sewage system, but disentangling these two factors is difficult. We also do not know why pH is so variable – there are any number of factors, from sewer pipe materials to wastewater inputs to sulfur oxidation, that could contribute the observed variation in pH across sites. As we do not know the specific mechanisms responsible for these differences in TSS and pH, we are hesitant to speculate further.

Section 3.2, paragraph 2: It is not clear to me how these 5 clusters were defined, e.g., what is the cut-off or criteria? In addition, do the site labels have meanings? Since there are clusters, I am curious if there are some common characteristics or patterns can be concluded? For example, do the residential buildings or teaching/lab buildings tend to cluster together? The discussion regarding the spatial variation is not sufficient in my opinion.

We provide details in the Methods section on how the clusters were identified. The clusters (identified by similarity in bacterial community composition) were not geographic clusters (i.e. sampling sites located in close proximity) – as is evident from Figure 1. Likewise, the clusters are not determined by building type as nearly all of the sewers sampled (15 of 17) served residential buildings. We have revised the text to clarify these points.

Section 3.3, paragraph 3: The observation of initial spikes of Legionella and Acinetobacter is very interesting. Would this be a health concern for the occupants? Was this also observed from other studies? If so, should an initial flushing of the stagnant systems be recommended for places like dormitories?

Great questions, but we do not want to speculate on these matters as this study was not designed to investigate these types of public health questions and there are other studies (cited here) that investigate these questions in more detail. Rather, we think this

is an important pattern worth highlighting given that our results are consistent with those reported elsewhere. We do not want to use our work to unnecessarily raise alarm about the risks of re-populating dormitory buildings.

Other comments:

Please add line numbers for easy of reviewing and commenting.

Figures are in general very blurry. Please fix this in future submission.

Changes made as recommended.

September 1, 2022

Dr. Noah Fierer
University of Colorado Boulder
Ecology and Evolutionary Biology
Boulder, CO

Re: mSystems00651-22R1 (A metagenomic investigation of spatial and temporal changes in sewage microbiomes across a university campus)

Dear Dr. Noah Fierer:

Your manuscript has been accepted, and I am forwarding it to the ASM Journals Department for publication. For your reference, ASM Journals' address is given below. Before it can be scheduled for publication, your manuscript will be checked by the mSystems production staff to make sure that all elements meet the technical requirements for publication. They will contact you if anything needs to be revised before copyediting and production can begin. Otherwise, you will be notified when your proofs are ready to be viewed.

Publication Fees:

If you would like to submit a potential Featured Image, please email a file and a short legend to mSystems@asmusa.org. Please note that we can only consider images that (i) the authors created or own and (ii) have not been previously published. By submitting, you agree that the image can be used under the same terms as the published article. File requirements: square dimensions (4" x 4"), 300 dpi resolution, RGB colorspace, TIF file format.

We recognize that the video files can become quite large, and so to avoid quality loss ASM suggests sending the video file via <https://www.wetransfer.com/>. When you have a final version of the video and the still ready to share, please send it to mSystems staff at mSystems@asmusa.org.

Sincerely,

Rachel Poretsky
Editor, mSystems

Journals Department
Fig. S1: Accept
Fig. S2: Accept
Fig. S5: Accept
Fig. S3: Accept
Fig. S4: Accept
Table S1: Accept